# Gender Differences in Atrial Fibrillation: From the Thromboembolic Risk to the Anticoagulant Treatment Response

**DOI:** 10.3390/medicina59020254

**Published:** 2023-01-28

**Authors:** Anna Rago, Ciro Pirozzi, Antonello D’Andrea, Pierpaolo Di Micco, Andrea Antonio Papa, Antonio D’Onofrio, Paolo Golino, Gerardo Nigro, Vincenzo Russo

**Affiliations:** 1Department of Cardiology, University of Campania “L.Vanvitelli”, Monaldi Hospital, 80131 Naples, Italy; 2Department of Cardiology, Umberto I Hospital, 84014 Nocera, Italy; 3Emergency Department, Rizzoli Hospital, Health Authority Naples 2, Ischia, 80076 Naples, Italy; 4Department of Cardiology, Monaldi Hospital, 80131 Naples, Italy

**Keywords:** atrial fibrillation, thromboembolic risk, female sex, anticoagulants, gender-differences

## Abstract

Atrial fibrillation (AF) is the most common cardiac arrhythmia associated with an increased thromboembolic risk. The impact of the female sex as an independent risk factor for thromboembolic events in AF is still debated. *Background and Objectives:* The aim of this review is to evaluate the gender-related differences in cardioembolic risk and response to anticoagulants among AF patients. *Materials and Methods:* The PubMed database is used to review the reports about gender differences and thromboembolic risk in atrial fibrillation. *Results:* Non-vitamin K oral anticoagulants (NOACs) represent the gold standard for thromboembolic risk prevention in patients with non-valvular atrial fibrillation (NVAF). Despite a similar rate of stroke and systemic embolism (SE) among men and women in NOACs or vitamin K antagonists (VKAs) treatment, the use of NOACs in AF women is associated with a lower risk of intracranial bleeding, major bleeding, and all-cause mortality than in men. *Conclusions:* The female sex can be defined as a stroke risk modifier rather than a stroke risk factor since it mainly increases the thromboembolic risk in the presence of other risk factors. Further studies about the efficacy and safety profile of NOACs according to sex are needed to support clinicians in performing the most appropriate and tailored anticoagulant therapy, either in male or female AF patients.

## 1. Introduction

Atrial fibrillation (AF) is the most common clinically relevant arrhythmia and increases the risk of both stroke and thromboembolism (TE) [1]. Women show lower AF prevalence compared to men; however, they present more severe symptomatology, worse quality of life, and a higher risk of stroke [2]. Several epidemiological studies have identified gender-related differences in the incidence of cardioembolic stroke and systemic embolism (SE) after correction for anticoagulation therapy, risk factors, age, and comorbidity in AF patients (Table 1 and Table 2). Non-vitamin K oral anticoagulants (NOACs) should be preferred over vitamin K antagonists (VKAs) for the prevention of stroke/SE among patients with non-valvular atrial fibrillation (NVAF). The efficacy and safety of NOACs have been demonstrated in large randomized trials [3,4,5,6] and confirmed in real-world settings among different clinical scenarios [7,8,9,10,11,12,13,14,15,16,17]. In particular, NOACs significantly reduce the combined endpoint of stroke/SE and intracranial hemorrhages (ICH). The gender-specific interactions related to responses to anticoagulant therapies have not been well understood. The aim of the present review is to evaluate the gender-related differences in cardioembolic risk and response to anticoagulants among AF patients.

## 2. Materials and Methods

The reports on gender differences and TE risk were reviewed through the PubMed database. Statistical gender differences in stroke rates were evaluated by examining the risk ratio and ***p***-value of various clinical trials. No external funding was used to support this work.

## 3. Results

### 3.1. Female Sex and Thromboembolic Risk Scores

The individual risk of systemic embolism and stroke in AF patients was defined by the retrospective analysis of large controlled clinical trials [18,34]. Since 2006, the CHADS2 score [congestive heart failure, hypertension, age ≥ 75 years, diabetes, stroke (doubled)] merged various stroke risk factors, and it has been indicated for the evaluation of thromboembolic risk in AF patients [35]. There is a relationship between the CHADS2 score and the stroke rate [36]; a CHADS2 score of zero identified AF patients at “low risk”, one to two at “moderate risk”, and greater than two at “high risk” for thromboembolic events. Nevertheless, AF patients with a CHADS2 score of zero showed an annual stroke rate > 1.5%, and it did not reliably identify those at truly low risk [37]. A limitation of the CHADS2 score was the lack of inclusion of some well-known stroke risk factors, such as vascular disease, female sex, and age ≥ 65 years [38]. Previous studies about TE risk in atrial fibrillation revealed that vascular disease, myocardial infarction, and peripheral artery disease are independent predictors of stroke and TE in AF patients [39,40,41]. Lip et al. [38] validated a novel risk factor-based approach to stroke risk stratification, “the Birmingham 2009 schema”, using the CHA2DS2-VASc acronym. In comparison with other published schema, the CHA2DS2-VASc score considered new risk factors (female gender, vascular disease, and age 65 to 74 years). According to this study’s results, age ≥ 75 years is a definitive high-risk factor; age between 65 and 74 years plus one additional risk factor is a condition in which anticoagulation therapy is necessary. The women enrolled in this study showed more thromboembolic events than men if they were not on anticoagulant therapy; therefore, the female sex has been included in the CHA2DS2-VASc score. Very few women were classified as intermediate risk through the CHA2DS2-VASc score (score of one = 15.1%) compared to CHADS2 score use (score of one = 34.9%). Additionally, there was a very low event rate in the CHA2DS2-VASc group having a score from zero to one (score of zero = 0%; score of one = 0.6%). CHA2DS2-VASc showed an increase in TE rate with increasing scores. Abraham et al. [42] concluded that CHADS2 is helpful for scores greater than or equal to two, but clinical decision-making about the prescription of anticoagulants becomes more ambiguous when the score is less than or equal to one because a score of CHADS2 equal to one places AF patients in an intermediate-risk class. Moreover, in these patients, the stroke risk doubles with every additional CHA2DS2-VASc point (female gender, vascular disease, and age 65 to 74 years). According to various studies [43,44,45], the CHA2DS2-VASc score showed a better performance in identifying patients at high and truly low thromboembolic risk, and it resulted in a refinement of TE risk stratification for AF. From 2010, the CHA2DS2-VASc score [congestive heart failure/left ventricular dysfunction, hypertension, age ≥ 75 (doubled), diabetes, stroke (doubled)—vascular disease, age 65–74, and sex category (female)] was proposed to better identify the “truly low-risk” patients who did not require any antithrombotic therapy, so that those with one or more stroke risk factors can be offered effective thromboprophylaxis [38]. The 2010 ESC Guidelines on AF management [37] recommended replacing the previous stratification of thromboembolic risk through CHADS2 score into low, moderate, and high risk with an approach based on “major” and “non-major clinically relevant” risk factors using the CHA2DS2-VASc score. The impact of the female sex as an independent risk factor for thromboembolic events in AF is still debated. Various studies showed an increased stroke and TE risk in AF women without antithrombotic and anticoagulant treatment, considering the female sex as an independent risk factor for thromboembolism [19,20,21,22]. Nevertheless, in a nationwide retrospective cohort study [23], women had a moderately increased risk of stroke compared with men with atrial fibrillation, but women younger than 65 years and without other risk factors did not show a significantly increased TE risk (0.7% vs. 0.5%, *p* = 0.09). Possible biases in some studies [19,23] are the older age and the presence of more stroke risk factors in enrolled women than men. Furthermore, in two studies [21,23], more men started anticoagulation therapy during the follow-up than women. Thus, the consequence of these biases may have overestimated the stroke/TE risk in AF women. Tsadok et al. [24] concluded that women, in particular over 75 years or older or in the presence of previous TE events, had a higher risk of stroke than men, regardless of their risk profile and use of anticoagulant treatment (adjusted HR 1.14, 95% CI [1.07, 1.22]; *p* < 0.001). Other studies [25,46] considered that the female sex increased the risk of stroke/TE only if associated with other risk factors, particularly in patients over 75 years. Therefore, the female sex should not be automatically considered as an independent stroke/thromboembolic risk factor and not be included in the CHA2DS2-VASc score in case of the age < 65 years and lone AF criterion. Based on these data, the female sex should be considered a “stroke risk modifier” rather than a “stroke risk factor” per se in AF [47]. Considering that the female gender alone does not confer an additional stroke risk, the CHA2DS2-VA score, with the exclusion of the female sex as a thromboembolic risk factor, has been recommended as an alternative to the CHA2DS2-VASc score in the 2018 Australian guidelines for atrial fibrillation [48]. Nielsen and colleagues [47] used the CHA2DS2-VA score for TE risk stratification in AF patients who had not initiated OAC therapy and reported that women had a higher CHA2DS2-VA score and higher stroke risk compared with men. Moreover, in the 5-year follow-up analysis, women exhibited a 23% higher stroke risk than men with the same CHA2DS2-VA score. Thus, this study confirmed that the female sex is a prognostic factor associated with AF-related stroke. In the presence of one or more non-sex stroke risk factors, women with AF consistently have significantly higher stroke risk than men, and OAC treatment should be considered. The potential consequences of recommending the sexless CHA2DS2-VA score could lead to the opinion that women carry less stroke risk than men and to a lower anticoagulant prescription in females [49]. According to the 2016 ESC Guidelines for the management of AF [50], anticoagulation is recommended with a single point or more “CHA2DS2-VASc” score for men and two or more for women. The 2020 European guidelines for the diagnosis and management of atrial fibrillation [51] confirmed the previous different oral anticoagulation recommendations based on gender and stressed the concept of female sex as a “stroke risk modifier” rather than a “stroke risk factor” per se in AF. Therefore, women with no other risk factors (CHA2DS2-VASc score of one) have a low stroke risk, similar to men with a CHA2DS2-VASc score of zero [23]. In contrast with the previously cited studies’ data, some authors found that the female sex was not a significant predictor of thromboembolic events/stroke in atrial fibrillation [27,28,29,30,31,32,52]. Possible bias in various studies could be the insufficient number of enrolled women to evaluate the correlation between the female sex and the risk of stroke in atrial fibrillation, explaining the absence of substantial differences between the sexes [27,28,32,52]. Other studies [29,31] did not show differences in stroke/TE events between the sexes, probably because the young study population had a low prevalence of stroke risk factors. In the EPICA study [30], the lack of gender difference could be explained by an increased stroke risk with increasing age in men, which equates the rates of ischemic events between males and females (Table 3).

### 3.2. Mortality Rate Gender Differences

The female sex is associated with higher mortality rates in the AF population across different socioeconomic regions. A meta-analysis of 30 studies, of which 5 randomized controlled trials (RCTs) and 24 observational studies, including a total of 4,371,714 participants, 66,511 with AF, showed a 12% higher risk of all-cause mortality, a 55% increased risk of cardiovascular (CV) mortality, and a two-fold increase in stroke risk associated with AF in women compared to men [54]. The Global Burden of Disease Study 2017 (GBD 2017) showed that AF global mortality was significantly lower in men at 2.87 (2.56–3.26) per 100,000 compared with women at 4.66 (4.99–4.83) per 100,000 population [55]. A Korean national cohort study showed higher mortality risk in female AF patients in all age groups in comparison to male AF patients [56]. The higher mortality causes in AF women are still unclear. Some previous studies showed that AF women tended to be undertreated with oral anticoagulants [57,58], even if the GLORIA-AF study did not show significant sex differences in anticoagulant prescription [59]. The increased thromboembolic risk of the female sex may lead to higher mortality [56]; however, there are contrasting data in the literature about the different mortality risks between men and women. In the ATRIA study [19], it was observed that the mortality related to ischemic stroke did not significantly differ by patient sex (23.4% for men and 23.7% for women; *p* = 0.94). Piccini et al. [26], in the observational cohort conducted through data from the ORBIT registry, revealed that women had lower all-cause mortality (adjusted HR 0.57, 95% CI [0.49, 0.67]) and cardiovascular death risk (adjusted hazard ratio 0.56, 95% CI [0.44, 0.72]) compared to men. Tsadok et al. [24] concluded that mortality rates were higher among men compared with women with AF (16.40, 95% CI [16.17, 16.62] vs. 13.74, 95% CI [13.55, 13.93]) and this difference persisted irrespective of age. In the observational cohort study by Potpara et al. [31], the rate of all-cause mortality, CV, and sudden death did not significantly differ between AF male and female patients. Salam et al. [32], in an observational study, showed that in-hospital mortality rates were comparable between AF women and men. Renoux et al. [33], in a cohort study including 147,622 AF patients, found that the mortality rate was 19% lower in women (RR 0.81, 95% CI [0.80, 0.83]) than in men at any age, including those over 75 years.

### 3.3. Biological Gender Differences and Thromboembolic Risk

Hypotheses about the greater predisposition to thromboembolic complications of atrial fibrillation in the female sex are often contrasting and need further investigations. A possible explanation of the gender differences in AF patients may arise from the low prevalence of women in each study (female sex prevalence about 33%) and a greater prevalence of women developing AF at older ages [60,61,62,63]. The advanced age predisposes to a worse prognosis and an increased probability of thromboembolic events, considering the presence of comorbidities (hypertension, diabetes mellitus, cardiomyopathies and heart failure, renal dysfunction, and prior stroke). Several studies reported that AF female patients have a greater stroke risk, because of a not properly prescribed oral anticoagulation; nevertheless, other studies deny these data. [28,62,64,65]. Furthermore, in various studies, women on VKAs seem to spend less time in the therapeutic range [66,67]; this hypothesis has been refuted by a comparative study conducted by Poli et al. [68]. Different biological features (endogenous and exogenous sex hormones, pharmacokinetics, and pharmacodynamics) between women and men could play an important role in the gender differences in AF thromboembolic risk (Table 4). Women more frequently present thyroid diseases, which may contribute to increased thromboembolic risk. Sheu et al. found that hyperthyroidism contributed to a 1.44-fold greater risk of ischemic stroke (95% CI adjusted hazard ratio 1.01 to 2.12; *p* = 0.038) [69]. In a study conducted on mice populations, Jayachandran et al. [70] demonstrated that beta estrogen platelet receptors are protective in oxidative stress, but they become deficient in post-menopausal mice. This results in a loss of platelet viability and an increased potential for the shedding of thrombogenic microparticles. The plasma concentrations of the inflammatory marker C-reactive protein [71] and the fibroblast growth factor-23 have been found to be higher in women than in men in various studies. Fibroblast growth factor-23 also has been associated with an increased risk of cardioembolic stroke [72]. Both factors have been associated with an increased risk of AF and might contribute to increased atrial fibrosis in women [73,74]. On the basis of the increased stroke risk in women and the different responses to anticoagulant therapy in patients with atrial fibrillation, there are structural differences in the myocardium and hormonal differences [75]. Structurally, the myocardium differs between the sexes. Men have a larger left atrium and increased ventricular wall thickness compared with women, partly explaining the higher prevalence of AF among men [76]. Advancing age and female sex are associated with a higher burden of atrial fibrosis, which contributes to thrombus formation and may explain the increased stroke risk in women [77]. Moreover, women seem to have a worse mechanical left atrial appendage function [78]. In postmenopausal women, there is a higher risk of arterial hypertension development than in men, especially from the seventh decade of life onwards [79]. Elevated arterial pressure is associated with arterial stiffness, characterized by collagen deposition, elastin degradation, and wall thickening with eventual dilatation [80]. Consequently, increased stiffness leads to an altered pulse wave velocity, which has been associated with increased stroke and CV mortality risks [81]. The women with AF present a greater degree of fibrosis, determining an increase in NT-proBNP and troponin I values associated with an increased risk of stroke, independently of the CHA2DS2-VASc score [82]. One biological explanation of the basis of the increased stroke risk in females relates to sex hormones. The expression of sex hormonal receptors in cardiomyocytes suggests that these hormones likely have a direct influence on electrophysiological processes and hence the propensity for AF development [83].

### 3.4. Gender Differences in VKAs Treatment Response

Fang et al. [19] concluded that women are at higher risk than men for AF-related thromboembolism off VKAs. Treatment with non-Vitamin K antagonists appears to be as effective in women, if not more so, than in men, with similar rates of major hemorrhage. Female sex, according to these authors, represents an independent risk factor for thromboembolism and should influence the decision to use anticoagulant therapy in the AF population. Senoo et al. [84] showed that anticoagulated female patients with atrial fibrillation had a similar rate of CV death and stroke/thromboembolic events but a lower risk of major bleeding compared with males. A meta-analysis of six randomized controlled trials conducted by Pancholy et al. [85] showed a greater residual risk of stroke/TE events in female patients with AF treated using VKAs prophylaxis compared with men (OR 1.279, 95% CI [1.111, 1.473], *p* = 0.001). This residual risk of stroke/TE events there was not in the population treated with NOACs. Sullivan et al. [86], in a randomized controlled trial, showed that women spent a greater percent time outside the therapeutic range (40 ± 0.7% in women vs. 37 ± 0.5% in men, *p* 0.001) or below (29 ± 0.7% in women vs. 26 ± 0.5% in men, *p* 0.0002) compared to men. Regardless, female AF patients who had a comparably high time spent in the therapeutic range (TTR > 66%) presented more stroke events (*p* 0.009) than men. The lower percent TTR in women compared with men could be explained by hormonal differences, care differences between men and women, different pharmacodynamics, and differences in the therapeutic efficacy of VKAs’ anticoagulation. Therefore, women might benefit from more frequent INR checks and require a higher therapeutic INR range to achieve a similar decrease in risk of ischemic stroke compared with men.

### 3.5. Gender Differences in Thromboembolic Risk and Anticoagulant Therapy in the Era of Direct Oral Anticoagulants

As recommended by the current guidelines [51] and shown by various real-life studies [87,88,89,90,91,92], NOACs are the preferred choice for thromboembolic risk prevention in NVAF patients, according to the CHA2DS2-VASc score. Anticoagulation is indicated with a point or more for men and two or more for women; therefore, anticoagulant treatment is not indicated for AF women without other risk factors [51]. It is known that gender differences in the efficacy and safety of antithrombotic treatment exist. The pathophysiological mechanisms underlying these gender differences are not fully understood and are likely multifactorial. The reasons for the known greater thromboembolic risk in women than in men remain unclear but may include older age, under-treatment with anticoagulant therapy, and poor anticoagulation control [93]. Many studies [94,95,96,97,98,99,100,101] were very well designed to assess the effectiveness of NOACs versus VKAs for thromboembolic risk prevention in NVAF, but sex-associated risk during NOACs treatment is not still understood. Law et al. [102] compared the effectiveness and safety outcomes of NOACs versus VKAs in 4972 men and 4834 women, showing that NOACs use was associated with a lower risk of intracranial hemorrhages (ICH) (HR: 0.16, 95% CI [0.06, 0.40]) and all-cause mortality (HR: 0.55, 95% CI [0.39, 0.77]) in women but not in men without significant differences in terms of stroke and TE incidence. In pivotal studies validating the efficacy and safety of NOACs versus VKAs, women were less represented than men [3,4,5,6]. Sex-specific differences in cardiovascular (CV) diseases could also impact the effects of CV therapies. The female sex was recognized as an independent predictor of thromboembolic risk, particularly in older patients. Interestingly NOACs, according to some authors’ data [103,104], seem to normalize the differences between males and females both in terms of safety and efficacy, whereas residual higher stroke risk and systemic embolism persist in AF women treated with vitamin K antagonist anticoagulants (VKAs) with optimal time in the therapeutic range. Based on the CHA2DS2-VASc score, NOACs represent the preferred choice in NVAF patients. Moreover, complete evaluation of apparently lower risk factors along with concomitant clinical conditions in AF patients appears mandatory, particularly for female patients, in order to achieve the most appropriate anticoagulant treatment, either in male or in female patients. The different outcomes of the various studies are likely due to the implementation of different studies (which, in part, had different comparators) and differences in the applied methodology. The following potential reasons on the basis of the gender differences in the anticoagulant treatment response in AF have been suggested [103]:
-Women are more frequently overdosed than men (probably due to differences in the volume of distribution and clearance of drugs);-They are more sensitive to drug effects than men (probably due to differences in receptor numbers, receptor binding, and the signal transduction pathway following receptor binding);-They take a greater amount of medications than men, with a higher potential for drug interactions.

Taken together, the available outcome results indicate small differences for men and women treated with NOACs. In AF, NOAC treatment in women results associated with similar overall efficacy but less major bleeding or ICH compared to males [104].

Various studies [105,106] observed that NOAC treatment might be insufficient for stroke prevention in AF patients with risk factors such as hypertension, smoking, dyslipidemia, diabetes, and advanced age, because of a potential residual risk of lacunar versus cardioembolic stroke, as these risk factors predispose to changes in the vascular bed and a prothrombotic state that NOACs cannot prevent.

A secondary analysis from the ENGAGE AF trial showed that the bleeding events were inferior in women treated with a 60 mg dose of edoxaban once daily versus warfarin in men. Stroke and TE events were similarly reduced between sexes. The exogenous factor Xa activity was 50% lower with a 30 mg dose of edoxaban once daily in comparison to a 60 mg dose of edoxaban with a slightly increased rate of ischemic events and reduced rate of bleeding events compared to VKAs use in both sexes. In this study, the use of 60 mg of edoxaban once daily in comparison with VKAs was more favorable in AF female patients than in male patients. This could be due to the increased number of women who qualified for a dose of 30 mg daily that were at higher risk of ischemic and bleeding events (annualized event rate 2.3% with a dose of 30 mg daily, versus 1.8% with a dose of 60 mg daily, versus 2% with VKAs assumption) [107]. Conversely, the secondary analyses from the other NOACs versus VKAs trials (ARISTOTLE trial, ROCKET AF trial, RE-LY trial) did not report differences in the safety and efficacy between the sexes [3,4,6,108,109].

In the GARFIELD AF trial, among the 28,624 enrolled patients (women 44.4%), women were more elderly (≥75 years) than men. It is well known that age above 65 years is associated with an increased risk of stroke. It is useful considering the impact of age on prescribing practice and adherence to anticoagulation therapy and on higher blood pressure (especially beyond menopause) in women than men. After adjustment for baseline risk factors in treated and untreated patients, HRs for stroke/SE rates were 1.3-fold higher in women (HR 1.30, CI [1.04, 1.63]) and similar for major bleeding (1.13, CI [0.85, 1.50]) and all-cause mortality (1.05, CI [0.92, 1.19]). The GARFIELD AF trial concluded that women have a higher stroke and TE risk, with an inferior reduction in stroke/TE rates during anticoagulation therapy than in men (HR 0.45, CI [0.33, 0.61] in men versus HR 0.77, CI [0.57, 1.03] in women) [110].

According to NOACs’ randomized controlled trials (RCT) gender differences subanalyses [107,108,109], the efficacy and safety benefits of NOACs were consistent in men and women, without significant gender differences in TE risk and mortality rate, compared with VKAs.

## 4. Conclusions

Gender-related differences in cardioembolic risk and response to anticoagulants among atrial fibrillation patients have been reported. Women have a lower prevalence of atrial fibrillation, are more symptomatic with a worse quality of life, and have an increased TE risk compared to men. The female sex can be defined as a “stroke risk modifier” rather than a “stroke risk factor” since it mainly increases the TE risk in the presence of other risk factors. NOACs represent the gold standard for TE risk prevention in patients with NVAF. Despite a similar rate of stroke and SE among men and women in NOACs or VKAs treatment, the use of NOACs in AF women is associated with a lower risk of intracranial bleeding, major bleeding, and all-cause mortality than in men. It is known that women frequently present concomitant modifiable risk factors, such as hypertension, thyroid dysfunction, obesity, or metabolic syndrome, which require a careful evaluation for appropriate thromboembolic risk stratification and anticoagulant treatment through a *tailored approach*, particularly in patients with low CHA2DS2-VASc score. Further studies about the efficacy and safety profile of NOACs according to sex are needed to support clinicians in performing the most appropriate and tailored anticoagulant therapy, either in male or female AF patients.

## Figures and Tables

**Table 1 medicina-59-00254-t001:** Comparison of data from studies of the association between the female sex and increased stroke/thromboembolism among AF patients.

Reference	Study	Study Type	No of AF Patients (% Women)	No of TE Events	Adjusted Risk (95% CI)	Mean Age [Years (SD)]	Conclusion
[18]	SPAF III	Observational cohort study	892 (22)	?	NS	67 (10)	
	SPAF I-III	Observational cohort study	2012 (28)	130	RR 1.6 (?), *p* = 0.01	69 (10)	Female patient > 75 years with AF are considered at high risk for stroke events
[19]	ATRIA	Comparative study	13,559	394	RR (1.6 (1.3–1.9) *p*-value is not reported	73.9% ≥ 70	Female sex is an independent risk factor for thromboembolism
[20]	The Framingham heart study	Observational cohort study	868 (47)	83	HR 1.92 (1.20–3.07) *p*-value is not reported	75(9)	Female sex is an independent risk factor for thromboembolism
[21]	The Copenhagen City Heart study	Observational cohort study	276 (39.9)	35 (22 in women,13 in men)	HR 2.6 (1.3–5.4) *p*-value is not reported	−58.5 (9.6) women without AF; −58.1 (9.7) man without AF; −69.0 (6.8) women with AF; −67.0 (8.4) men with AF	AF predisposes to a 4-fold increased risk of stroke and to a 2.5-fold increased risk of cardiovascular death
[22]	Tze-Fan Chao et al.	Observational cohort study	829 (38.6)	14 in women	HR 7.77 (3.97–15.19; *p* = 0.001)	45.4 (12.0)	Female sex is a risk factor for stroke, in fact women with AF and CHA2DS2-VASc score 1 (attributable to gender) are more predisposed to stroke events.
[23]	Friberg L et al.	Nationwide cohort study	100,802 (50.3)	7221 (59% in women)	HR 1.18 (1.12 to 1.24) *p* < 0.001	Male mean age 74.7 (13.5) Female mean age 80.9 (10.6)	Women with AF have an incremented risk of stroke compared with men but women < 65 years without other risk factors are considered at low risk and do not need anticoagulation
[24]	Avgil Tsadok M., et al.	Observational cohort study	83,513 (52.8)	4266 (2570 in women and 1696 in men)	HR, 1.14 (1.07–1.22); *p* < 0.001	Women 80.2 (74.8–5.4) Men 77.2 (72.2–2.4)	The stroke risk is greater in women >75 years than in men, regardless of VKAs use
[25]	Mikkelsen et al.	Nationwide cohort study	87,202 (51.3)	1-year 5452 (59.8% in women) 12-year 12,962 (58.4% in women)	1-year follow-up age: < 65: 0.89 (0.70–1.13) 65–74: 0.91 (0.79–1.05) ≥ 75: 1.20 (1.12–1.28) 12-year follow-up age: < 65: 0.86 (0.76–0.98) 65–74: 0.98 (0.90–1.07) ≥ 75: 1.10 (1.05–1.15)	Male 71.0 ± 14.3 Female 78.2 ± 12.1	Female sex is only associated with an increased risk of stroke for AF patients aged ≥ 75 years
[26]	Jonathan P. Piccini et al.	Observational cohort study	10,135 (42)	−118 in women −96 in men	HR 1.39 (1.05–1.84; *p* = 0.02)	−77 (69–83) women −73 (65–80) men	Women have lower risk-adjusted all-cause and cardiovascular death compared with men, but higher stroke rates.

CI—confidence interval; HR—hazard ratio; RR—relative risk; OR—odds ratio; NS—not significant; SD—standard deviation; ?—not reported; SPAF—stroke prevention in atrial fibrillation; ATRIA—anticoagulation and risk factors in atrial fibrillation.

**Table 2 medicina-59-00254-t002:** Comparison of data from studies not reporting gender differences in stroke/thromboembolism risk.

Reference	Study	Study Type	No of AF Patients (% Women)	No of TE Events	Adjusted Risk (95% CI)	Mean Age [Years (SD)]	Conclusion
[27]	Ralph F. Bosch	Randomized Controlled Trial	2742 patients (37.2% women, mean age 71.2 years	−95 (58 in men −37 in women)	OR 0.97 (0.78–1.21) *p* 0.79	−67.5 men −71.2 women	Outcomes (stoke included) do not differ much between the sexes
[28]	Hiroshi Inoue et al.	Prospective multicenter study	5241 male subjects, 2165 female subjects	−92 in men −34 in women	OR 1.12 (0.75–1.67) *p* 0.576 in univariate analysis	70	Female gender is not a risk factor for thromboembolic events
[29]	Lin et al.	Nationwide database analysis	7920 patients (45.9)	329	OR 0.942 (0.787–1.127) *p*-value 0.512	Subjects aged 20–64 years, 30.9% aged 65–74 years and 32.4% aged over 75 years.	The gender is not associated with ischemic stroke
[30]	Poli et al.	Comparative Study	3015 (54.9)	591 (19.6): −255 in men −336 in women	RR 1.2 (0.8–1.8) *p*-value 0.25	83 (80–102)	Absence of significant differences in the risk of major clinical outcomes between genders, in- cluding mortality, stroke
[31]	Potpara et al.	Observational cohort study	862 (36.5)	−14 in female patients −26 in male patients	RR 0.9 (0.5–1.7) *p* 0.771	52.2 (12.1)	Thromboembolism and stroke events do not differ between the sexes
[32]	Amar M Salam et al.	Observational Study	3849 (36.8)	15	?	−54.5 (15.7) men −59 (15) women	Stroke rates are comparable between the sexes
[33]	Renoux et al.	Cohort study	147,622 (51.8)	11,326	RR 1.01 (0.97, 1.05) *p*-value is not reported	−73.7 (11.2) men −77.1 (11.2) women	Women are not at higher risk of thromboembolic events than men.

CI—confidence interval; RR—relative risk; OR—odds ratio; SD—standard deviation; ?—not reported.

**Table 3 medicina-59-00254-t003:** Thromboembolic risk scores reported in the literature.

Scores	Thromboembolic Risk Factors—(Points)	Score	Adjusted Stroke Rate %/Year	References
CHADS2 score	Congestive heart failure or LV dysfunction (1) Hypertension (1) Age ≥ 75 years (1) Diabetes mellitus (1) Stroke or TIA or systemic thromboembolism (2)	Maximum total score: 6 0 1 2 3 4 5 6	1.9 2.8 4.0 5.9 8.5 12.5 18.2	[36]
CHA2DS2-VASc score	Congestive heart failure or LV dysfunction or HCM (1) Hypertension (1) Age ≥ 75 years (2) Diabetes mellitus (1) Stroke or TIA or systemic thromboembolism (2) Vascular disease (prior myocardial infarction or peripheral arterial disease or complex aortic atheroma or plaque on imaging) (1) Age 65–74 (1) Sex category (female) (1)	Maximum total score: 9 0 1 2 3 4 5 6 7 8 9	0 1.3 2.2 3.2 4 6.7 9.8 9.6 6.7 15.2	[38]
CHA2DS2-VA score	Congestive heart failure or LV dysfunction (1) Hhypertension (1) Age ≥ 75 years (2) Diabetes mellitus (1) Stroke or TIA or systemic thromboembolism (2) Vascular disease (prior myocardial infarction or peripheral arterial disease or complex aortic atheroma or plaque on imaging) (1) age 65–74 (1)	Maximum total score: 8 0 1 2 ≥3	0.72 1.09 2.44 5.46	[48,53]

LV—left ventricle; HCM—hypertrophic cardiomyopathy; TIA—transient ischemic attack.

**Table 4 medicina-59-00254-t004:** Biological features predisposing to greater stroke risk in AF women than men.

Biological Features	Physiological Mechanisms
AF development at older age in women	AF diagnosis at advanced age is associated with the presence of more stroke risk factors (hypertension; diabetes mellitus, cardiomyopathies and heart failure, renal disfunction, prior stroke).
Hypertension	Increased stiffness leads to an altered pulse wave velocity, that has been associated with increased stroke and CV mortality risks.
Time spent in therapeutic range (TTR) during VKAs treatment	Women on VKAs seem to spend less time in therapeutic range. The lower percent TTR in women compared with men could be explained by hormonal differences, care differences between men and women, different pharmacodynamics, and therapeutic efficacy of VKAs anticoagulation.
Hormonal differences	Beta estrogen platelet receptors are protective in oxidative stress, but they become deficient in post-menopausal age. This results in a loss of platelet viability and increased potential for shedding of thrombogenic microparticles. Hormones likely have a direct influence on electrophysiological processes and hence the propensity for AF development.
Different pharmacokinetics and pharmacodynamics among men and women	Different pharmacodynamics and therapeutic efficacy of anticoagulation
Thyroid diseases	The mechanisms of association between hyperthyroidism and stroke are not fully understood. Hyperthyroidism is associated with hypertension, hypercoagulable state, and endothelial dysfunctions, which could increase stroke risk.
Proinflammatory state	The plasma concentrations of the inflammatory marker C-reactive protein and of fibroblast growth factor-23 have been found to be higher in women than in men. These factors are associated with increased risk of cardioembolic stroke.
Different myocardium structure in women (higher burden of atrial fibrosis)	A worse mechanical left atrial appendage function, hormones and an increased proinflammatory state in women. predispose to a higher burden of atrial fibrosis determining stroke events.

## Data Availability

No new data were created or analyzed in this study. Data sharing does not apply to this article.

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
