# Peer review of "Gender Differences in Atrial Fibrillation: From the Thromboembolic Risk to the Anticoagulant Treatment Response"

_medicina, 2023, doi:10.3390/medicina59020254_

Round 1
Reviewer 1 Report
The aim of the present review was to evaluate the gender-related differences in cardioembolic risk and response to anticoagulants among AF patients. Non vitamin K oral anticoagulants (NOACs) represent the gold standard for the TE risk prevention in patients with NVAF. De-spite a similar rate of stroke and SE among men and women in NOACs or Vitamin K Antagonists (VKAs) treatment, the use of NOACs in AF women is associated with a lower risk of intracranial bleeding, major bleeding and all-cause than mortality in men. Female sex can be defined as a stroke risk modifier; Rather than a stroke risk factor, since it mainly increases the TE risk in the presence of other risk factors.
The review is well written, but it does not include new perspectives on the influence of female sex on mortality or risk of atrial fibrillation, e.g. Okumura et al, Wańkowicz et al.
Author Response
We thank the reviewer for the comment that allows us to improve the manuscript.. We cited Okumura et al and Wańkowicz et al. studies (Results- 3.5 paragraph, lines 317-321), as you suggested.

Reviewer 2 Report
While the topic is very interesting and relevant to the field, it requires extensive English and structure formatting.
It does not follow a smooth flow and the ideas are not well connected. Overall is difficult to extract the final message, even is a controversial topic and relevant studies were reviewed the presentation is confusing for the reader.
I strongly suggest using more tables and graphs to summarize the information, as well as organizing the discussion first all the studies that support gender differences then discuss the contra part and comment the possible reasons of the differences on the results rather than mixing it together.
Table 1 is missing some p values. in the same table should include a column that summarize the conclusions of each study. the table also will benefit if separates the studies that found gender differences and the ones that didn't
Please include a graphical summary of the biological sex differences for thrombotic risk
The discussion of different scores on the text is long and confusing. A table summarizing the scores should be included instead.
There are no references supporting authors claims regarding anticoagulant treatment response (line 314)
Additionally, a discussion of the gender results of big NOACS RCT is missing.
Author Response
We thank the reviewer for the comment that allows us to improve the manuscript.
- We provided extensive English and structure formatting as you suggested.
- We reviewed the presentation trying to improve the structure of the manuscript about a controversial topic and we added more tables and graphs, as you suggested
- We added in table 1 the missing reported p values and included a column summarinzing the conclusions of each study. We also separated the studies that found gender differences and the ones that didn't (table 1,table2).
-We added table 3 summarizing the thromboembolic risk scores (Results, 3.1 paragraph).
-We included a graphical summary of the biological sex differences for thrombotic risk in the table 4 (Results- 3.3 paragraph).
-The reference supporting authors claims regarding anticoagulant treatment response is number 107 (cited in Results- 3.5 Paragraph, Line 306).
- We adde discussion of the gender results of big NOACS RCT (Results-3.5 paragraph, Lines 322-324), as you suggested

Round 2
Reviewer 1 Report
I accept this review for publication without corrections.
Author Response
We thank the reviewer for the comment
Reviewer 2 Report
1.While the authors provide a table (table 3) with the summary of the scores, it would be important to include the values of each item for the scores and the classification and interpretation of the score so the differences can be appreciated easily
2.The authors did not provided a graphical summary (figure) rather provide a table (table 4) with a list of predisposing factors for greater risk in women. if the authors insist in using a table please provide a new column with a brief but detailed explanation of the physiological mechanisms of each factor to stroke risk, so the reader can understand the role of the items in the thrombotic risk
3. References of sex differences of big RCT were provided, however the discussion about them seems limited, considering the importance of these trials in clinical use of anticoagulation I consider they deserve a more detailed discussion and/or a table to summarize the results, please include information from ROCKET-AF, GARFIELD-AF, ENGAGE-AF, ARISTOTLE
Author Response
Dear Editor,
we sincerely appreciate the consideration of our manuscript and we gratefully thank the reviewers
for their interesting comments. We improved the manuscript clarifying the unclear aspects. Below
you’ll find the point-to-point response to the reviewers’ comments.
Major concerns: 1. While the authors provide a table (table 3) with the summary of the scores, it would be important to include the values of each item for the scores and the classification and interpretation of the score so the differences can be appreciated easily
We thank the reviewer for the comment that allows us to improve the manuscript.
We include the values of each item for the scores and the classification and interpretation of the score in the table 3(Results-3.1 paragraph), as you suggested.
Table 3. Thromboembolic risk scores reported the literature
Scores |
Thromboembolic risk factors |
Score |
Adjusted Stroke Rate %/year |
References |
CHADS2 score |
Congestive heart failure or LV dysfunction, Hypertension, Age ≥75 years, Diabetes Mellitus, Stroke or TIA or systemic thromboembolism |
Maximum total score: 6 0 1 2 3 4 5 6 |
1.9 2.8 4.0 5.9 8.5 12.5 18.2 |
[22] |
CHA2DS2-VASc score |
Congestive heart failure or LV dysfunction or HCM, Hypertension, Age ≥75 years, Diabetes mellitus, Stroke or TIA or systemic thromboembolism, Vascular disease (prior myocardial infarction or peripheral arterial disease or complex aortic atheroma or plaque on imaging), Age 65-74, Sex category (female) |
Maximum total score: 9 0 1 2 3 4 5 6 7 8 9 |
0 1.3 2.2 3.2 4 6.7 9.8 9.6 6.7 15.2 |
[24] |
CHA2DS2-VA score |
Congestive heart Failure or LV dysfunction, Hypertension, Age ≥75 years, Diabetes Mellitus, Stroke or TIA or systemic thromboembolism, Vascular disease (prior myocardial infarction or peripheral arterial disease or complex aortic atheroma or plaque on imaging), Age 65-74 |
Maximum total score: 8 0 1 2 ≥3
|
0,72 1.09 2,44 5,46 |
[41] [115] |
LV: left ventricle; HCM: hypertrophic cardiomyopathy; TIA: transient ischemic attack
2.The authors did not provided a graphical summary (figure) rather provide a table (table 4) with a list of predisposing factors for greater risk in women. if the authors insist in using a table please provide a new column with a brief but detailed explanation of the physiological mechanisms of each factor to stroke risk, so the reader can understand the role of the items in the thrombotic risk
We thank the reviewer for the comment that allows us to improve the manuscript.
We provided a new column with a brief but detailed explanation of the physiological mechanisms of each factor to stroke risk in the table 4(Results-3.3 paragraph), as you suggested
Table 4. Biological features predisposing to greater stroke risk in AF women than men
AF development at older age in women |
AF diagnosis at advanced age is associated with the presence of more stroke risk factors (hypertension; diabetes mellitus, cardiomyopathies and heart failure, renal disfunction, prior stroke). |
Hypertension |
Increased stiffness leads to an altered pulse wave velocity, that has been associated with increased stroke and CV mortality risks. |
Time spent in therapeutic range (TTR) during VKAs treatment |
Women on VKAs seem to spend less time in therapeutic range. The lower percent TTR in women compared with men could be explained by hormonal differences, care differences between men and women, different pharmacodynamics and therapeutic efficacy of VKAs anticoagulation. |
Hormonal differences |
Beta estrogen platelet receptors are protective in oxidative stress but they become deficient in post-menopausal age. This results in a loss of platelet viability and increased potential for shedding of thrombogenic microparticles. Hormones likely have a direct influence on electrophysiological processes and hence the propensity for AF development. |
Different pharmacokinetics and pharmacodynamics among men and women |
Different pharmacodynamics and therapeutic efficacy of anticoagulation |
Thyroid diseases |
The mechanisms of association between hyperthyroidism and stroke are not fully understood. Hyperthyroidism is associated with hypertension, hypercoagulable state and endothelial dysfunctions, which could increase stroke risk. |
Proinflammatory state |
The plasma concentrations of the inflammatory marker C-reactive protein and of fibroblast growth factor-23 have been found to be higher in women than in men. These factors are associated with increased risk of cardioembolic stroke. |
Different myocardium structure in women (higher burden of atrial fibrosis) |
A worse mechanical left atrial appendage function, hormones and an increased proinflammatory state in women. predispose to a higher burden of atrial fibrosis determining stroke events. |
- References of sex differences of big RCT were provided, however the discussion about them seems limited, considering the importance of these trials in clinical use of anticoagulation I consider they deserve a more detailed discussion and/or a table to summarize the results, please include information from ROCKET-AF, GARFIELD-AF, ENGAGE-AF, ARISTOTLE
We thank the reviewer for the comment that allows us to improve the manuscript.
We added a more detailed discussion to summarize the results of ROCKET-AF, GARFIELD-AF, ENGAGE-AF, ARISTOTLE (Results – 3.5 paragraph, lines 312-335), as you suggested
A secondary analysis from ENGAGE AF trial showed that the bleeding events were inferior in women treated with dose of edoxaban of 60 mg once daily versus VKAs than in men. Stroke and TE events were similarly reduced between sexes. The exogenous factor Xa activity were 50% lower with a dose of edoxaban of 30 mg once daily in comparison to dose of edoxaban 60 mg with a slightly increased rate of ischemic events and reduced rate of bleeding events compared to VKAs use in both of sexes. In this study the use of edoxaban 60 mg once daily in comparison with VKAs was more favorable in AF female patients than in male patients. This could be due to increased number of women qualified for dose of 30 mg daily, that were at higher risk of ischemic and bleeding events (annualized event rate 2.3% with a dose of 30 mg daily, versus 1.8 % with a dose of 60 mg daily, versus 2% with VKAs assumption) [111]. Conversely, the secondary analyses from the other NOACs versus VKAs trials (ARISTOTLE trial, ROCKET AF trial, RE-LY trial) did not report differences in the safety and efficacy between the sexes [3,4,6,112,113].
In the GARFIELD AF trial among the 28624 enrolled patients (women 44.4%), women were more elderly (≥75 years) than men. It is well known that age above 65 years is associated with an increased risk of stroke. It is useful considering the impact of age on prescribing practice and adherence to anticoagulation therapy and on higher blood pressure (especially beyond menopause) in women than men. After adjustment for baseline risk factors in treated and untreated patients, HRs for stroke/SE rates were 1.3-fold higher in women (HR 1.30 (1.04 to 1.63)), and similar for major bleeding (1.13 (0.85 to 1.50)) and all-cause mortality (1.05 (0.92 to 1.19)).The GARFIELD AF trial concluded that women have a higher stroke and TE risk with an inferior reduction in stroke/TE rates during anticoagulation therapy than in men [(HR 0.45 (0.33 to 0.61) in men versus HR 0.77 (0.57 to 1.03) in women)] [114].
According to NOACs randomized controlled trials (RCT) gender differences subanalyses [111-113], the efficacy and safety benefits of NOACs were consistent in men and women, without significant gender differences in TE risk and mortality rate, compared with VKAs.
